# Can Inositol Pyrophosphates Inform Strategies for Developing Low Phytate Crops?

**DOI:** 10.3390/plants9010115

**Published:** 2020-01-17

**Authors:** Catherine Freed, Olusegun Adepoju, Glenda Gillaspy

**Affiliations:** Department of Biochemistry, Virginia Tech, Blacksburg, VA 24061, USAade1704@vt.edu (O.A.)

**Keywords:** inositol, inositol phosphate, inositol pyrophosphate, inositol phosphate signaling, inositol phosphate kinases, PPIP5K, ITPK, phytate

## Abstract

Inositol pyrophosphates (PP-InsPs) are an emerging class of “high-energy” intracellular signaling molecules, containing one or two diphosphate groups attached to an inositol ring, that are connected with phosphate sensing, jasmonate signaling, and inositol hexakisphosphate (InsP_6_) storage in plants. While information regarding this new class of signaling molecules in plants is scarce, the enzymes responsible for their synthesis have recently been elucidated. This review focuses on InsP_6_ synthesis and its conversion into PP-InsPs, containing seven and eight phosphate groups (InsP_7_ and InsP_8_). These steps involve two types of enzymes: the ITPKs that phosphorylate InsP_6_ to InsP_7_, and the PPIP5Ks that phosphorylate InsP_7_ to InsP_8_. This review also considers the potential roles of PP-InsPs in plant hormone and inorganic phosphate (P*i*) signaling, along with an emerging role in bioenergetic homeostasis. PP-InsP synthesis and signaling are important for plant breeders to consider when developing strategies that reduce InsP_6_ in plants, as this will likely also reduce PP-InsPs. Thus, this review is primarily intended to bridge the gap between the basic science aspects of PP-InsP synthesis/signaling and breeding/engineering strategies to fortify foods by reducing InsP_6_.

## 1. Introduction

The inositol phosphate (InsP) signaling pathway has been implicated in a diverse array of cellular processes across eukaryotic organisms (for reviews, see [1,2,3]). Inositol phosphates are intracellular signaling molecules built around a simple 6-carbon *myo*-inositol (inositol) scaffold used in signal transduction (Figure 1). Differing phosphorylation patterns on the inositol ring convey specific cellular information, and several combinations of phosphorylation events are possible. Inositol hexakisphosphate (InsP_6_), also called phytate when complexed with cations, is a molecule where all hydroxyl (OH) groups on the inositol ring are phosphorylated, that serves as both a storage pool of phosphate as well as a signaling molecule in plants [4,5,6].

InsP_6_ is the major phosphorus storage sink within the plant seed, comprising up to ~1% of an Arabidopsis seed’s dry weight, and roughly 65–85% of total seed phosphorus in cereal crops [6]. InsP_6_ is one of the most highly electronegative molecules present in the cell, resulting in a greatly limited ability to cross cell membranes. It can be transported into the vacuole by MRP5, an ABC-transporter localized in the vacuolar membrane that has been shown to specifically transport InsP_6_ [4]. During seed development, the electronegative properties of InsP_6_ result in the chelation of positively charged metal ions such as Mg^2+^, Fe^2+^, Zn^2+^, Mn^2+^, Ca^2+^, and these InsP_6_ metal complexes accumulate within the protein storage vacuole (PSV) (Figure 2) [7,8].

While InsP_6_ is important for plant survivorship and yield, it is also, unfortunately, an anti-nutrient, as it can prevent useful cations and minerals from being absorbed by the animal gut [9,10,11]. Additionally, non-ruminant animals cannot digest InsP_6_, and that which is excreted from livestock is a major environmental concern as it leads to phosphorus pollution, eutrophication, and toxified watersheds [12,13]. Given this, plant breeders and biotechnologists have sought to limit the production of InsP_6_ in plants, resulting in the so-called low-phytate crops [14]. However, reducing InsP_6_ synthesis and accumulation in plants can come with consequences, such as a significant decline in vital plant signaling molecules derived from InsP_6_.

**Figure 2 plants-09-00115-f002:**
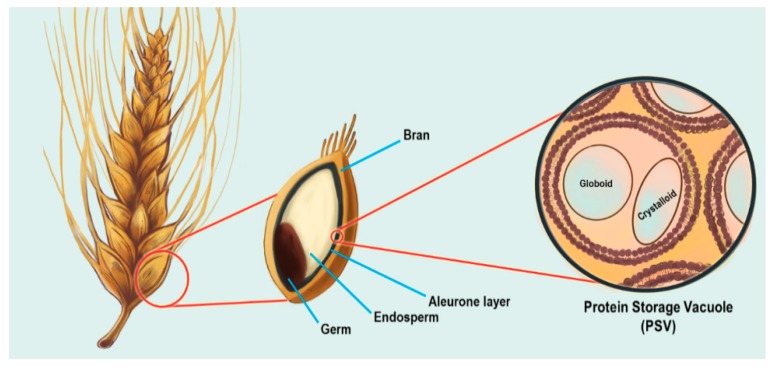
Diagram illustrating phytate storage in wheat granules. Phytate and other cations are stored in the protein storage vacuole (PSV) within small, crystalline storage bodies, known as globoids [9,15]. While the location of globoids varies across species, a majority of these are found within the aleurone layer of cereal crops, such as wheat and barley, as well as oilseed crops including peanuts, hemp, sunflower, and cotton [6,9,16]. Maize globoids are concentrated within the germ, whereas soybean phytate is distributed throughout the entire seed [6,17,18]. PSVs also contain crystalloid bodies, containing stored proteins and lipids [9].

Our primary expertise and focus are on an emerging class of InsPs derived from InsP_6_ that contain one or two diphosphate or pyrophosphate groups (PP) attached to the inositol ring (Figure 1) [5,19]. These inositol pyrophosphates (PP-InsPs) are gaining significant attention due to their newly discovered roles in energetic metabolism as well as hormone signaling and P*i* sensing (for a review, see [20]). The presence of pyrophosphate bonds in PP-InsPs has resulted in their consideration as “energetic signaling” molecules, and we note that PP-InsPs are similar in structure to ADP or ATP. PP-InsP biosynthesis is well described in non-plant eukaryotes, such as yeast and mammals, and many physiological roles have been linked to these molecules [1,2,3]. Elucidating the route of PP-InsP biosynthesis in plants has recently been accomplished and is fundamentally critical to our understanding of these molecules, which is described in the following sections. We will start with an overview of how plants synthesize the precursor to PP-InsPs, InsP_6_, as several of the enzymes in this pathway are key to understanding the PP-InsP pathway.

## 2. InsP_6_ Synthesis: The Lipid-Dependent Pathway

InsP_6_ can be synthesized by two interconnected pathways in plants. The pathways are named for their starting material: the Lipid-Dependent pathway and the Lipid-Independent pathway (Figure 3). The Lipid-Dependent pathway is present in all eukaryotic organisms [21,22,23,24,25]. The Lipid-Independent pathway for synthesizing InsP_6_ was originally discovered in *Dictyostelium* in a landmark paper by Stephens and Irvine in 1990, and followed up on in *Spirodela polyrhiza* [26,27]. This pathway was thought be unique to these organisms, along with land plants. However, a very recent publication by Desfougères et al. shows that the Lipid-Independent pathway is also present in mammals [28]. This work is the first to report evidence of the Lipid-Independent pathway in mammals and will be crucial for exploring the evolution of enzymes across organisms.

The lipid component in the Lipid-Dependent pathway is phosphatidylinositol phosphate (PtdInsP), a molecule containing inositol as the head group (Figure 1). While plants synthesize a myriad of lipid-soluble PtdInsPs, phosphatidylinositol (4,5) bisphosphate (PtdIns(4,5)P_2_), is important for the Lipid-Dependent pathway as it is acted on by the enzyme phospholipase C (PLC) [29]. Phospholipases, by definition, hydrolyze phospholipids. The hydrolysis of PtdIns(4,5)P_2_ by PLC produces Ins(1,4,5)P_3_ and diacylglycerol (DAG), which essentially converts a phosphorylated lipid-signaling molecule (PtdIns(4,5)P_2_) into a water-soluble, InsP-signaling molecule (Ins(1,4,5)P_3_) (Figure 3) [29].

Ins(1,4,5)P_3_ can be subsequently phosphorylated into InsP_4_, and then InsP_5_, by a dual specific inositol polyphosphate multikinase (IPMK). The IPMK enzyme is encoded by two genes in Arabidopsis, *AtIPK2α* and *AtIPK2β* (Table 1) [30]. Both genes encode enzymes with a 6/3-kinase activity, catalyzing the conversion of Ins(1,4,5)P_3_ to Ins(1,4,5,6)P_4_ and to a final Ins(1,3,4,5,6)P_5_ product [31]. 5-kinase activity towards Ins(1,3,4,6)P_4_ and Ins(1,2,3,4,6)P_5_ was also reported by Stevenson-Paulik et al. Notably, these genetic studies show that the *AtIPK2β* gene can complement a yeast *ipk2* mutant, and that Arabidopsis T-DNA loss-of-function *atipk2β* mutants have a 35% reduction in seed InsP_6_ (Table 1) [30]. *atipk2α* mutants are not easily studied as those that are recovered are lethal. This, along with the generally ubiquitous expression of *AtIPK2α*, supports the idea that *AtIPK2α* supplies the major IPK2 or IPMK activity in the plant cell.

The last step in the Lipid-Dependent pathway of InsP_6_ biosynthesis is the phosphorylation of Ins(1,3,4,5,6)P_5_ to InsP_6._ This step is catalyzed by only one type of enzyme in nature, the inositol pentakisphosphate 2-kinase (IPK1), so named because all known IPK1 enzymes phosphorylate the 2-position of InsP_5_ [30]. While there are seven genes in Arabidopsis that are predicted to encode IPK1 enzymes, the *At5g42810* gene (*AtIPK1*) is the only one actively expressed in plants [30]. Complementation assays reveal that *AtIPK1* is able to complement a yeast *ipk1* mutant, restore InsP_6_ levels, and rescue the mutant’s temperature-sensitive growth phenotype [39]. As loss of IPK1 function results in an 83% reduction in InsP_6_ in seeds (Table 1), this shows that IPK1 plays a major role in maintaining seed InsP_6_ levels [30].

## 3. InsP_6_ Synthesis: The Lipid-Independent Pathway

Given the importance of P*i* storage in plants, it is not surprising that plants evolved a separate way to synthesize InsP_6_, apart from the PtdInsP pathway. In the Lipid-Independent pathway, “free” *myo-*inositol is acted on by a series of inositol kinases. The first is the *myo-*inositol kinase (MIK), which was first identified in maize as a product of the *Low Phytic Acid* gene [40]. Loss-of-function maize *lpa3* mutants have reduced InsP_6_ and elevated inositol levels in the seeds [40]. Arabidopsis *atmik* mutants have a large reduction in total seed mass InsP_6_ levels (Table 1) [35]. The second step in the Lipid-Independent pathway is likely catalyzed by a gene/protein named LPA1 in rice [41]. This gene product was originally categorized as a potential 2-phosphoglycerate kinase that impacts InsP_6_ accumulation [41]. However, recent structural modeling indicates that InsP_3_ can be accommodated in the active site of this kinase, supporting a role for it as an InsP kinase [42].

The next step in this pathway should involve an inositol kinase capable of phosphorylating an InsP_2_ substrate. As no such kinase has been identified, this prompted speculation that InsP_2’_s conversion to InsP_3_ is catalyzed by a moonlighting function of another inositol phosphate kinase, which has yet to be identified. The last novel component of the Lipid-Independent pathway is the inositol triphosphate kinase (ITPK), which can phosphorylate specific InsP_3_ and InsP_4_ molecules [39,43]. A gene encoding ITPK1 was also identified in a maize mutant named *lpa2* [44]. There are four ITPK enzymes (AtITPK1–4) in Arabidopsis [45]. It is interesting to note that only Arabidopsis *atitpk1* and *atitpk4* mutants have reduced seed InsP_6_ levels (Table 1) [33,35]. Our own work with AtITPK1 and AtITPK2 enzymes shows that these proteins are also efficient at converting InsP_6_ to InsP_7_ [38]. Laha et al. additionally used NMR to show that 5PP-InsP_5_ is the isoform synthesized by AtITPK1 and AtITPK2 [36]. These are key findings that highlight the catalytic flexibility of the ITPKs, as well as indicating that the Lipid-Independent pathway may have an important relationship with, and impact on, PP-InsP synthesis.

The ITPKs are thought to act in concert with the IPK2 enzymes in producing Ins(1,3,4,5,6)P_5_ in the Lipid-Independent pathway [27,31,43]. The final step is the conversion of Ins(1,3,4,5,6)P_5_ to InsP_6_. Both pathways utilize IPK1 to synthesize InsP_6_ and converge at this last step in InsP synthesis. This further highlights the importance of IPK1 in InsP_6_ synthesis. A very recent publication on non-plant organisms suggests that there might be some conserved functions of the Lipid-Independent pathway in other eukaryotes. Humans, for example, contain ITPK genes, and the expression of these enzymes appears to complement mutants defective in PLC, which cannot synthesize InsP_6_ [28,46]. Although more work needs to be done, this suggests that animals may also utilize ITPKs in a Lipid-Independent InsP_6_ pathway. In the same work, these authors also found that the plant ITPK1 could complement the yeast PLC mutant, suggesting that the plant ITPK may also have a very flexible substrate preference, and may act at several different steps in the Lipid-Independent pathway [28].

## 4. PP-InsP Synthetic Pathway

While a great majority of InsP_6_ is stored as phytate, a small pool can be further phosphorylated to form PP-InsPs. Our group and others have examined a class of enzymes involved in PP-InsP synthesis named the diphosphoinositol-pentakisphosphate kinases (PPIP5Ks), known as VIP or VIH in plants, and Vip1 in *Chlamydomonas reinhardtii* (algae) [5,34,47]. Mammalian and yeast PPIP5K enzymes phosphorylate the 1-position on InsP_6_ and 5PP-InsP_5_ to generate an InsP_7_ molecule, 1PP-InsP_5_, and an InsP_8_ molecule, 1,5(PP)_2_-InsP_4_, respectively [48,49,50,51,52]. Two Arabidopsis genes, *AtVIP1* and *AtVIP2*, are orthologous to the mammalian *PPIP5K* genes [5,34]. AtVIP1 (also referred to as AtVIH2) and AtVIP2 (AtVIH1) are dual-domain enzymes, consisting of an ATP-grasp N-terminal kinase domain (KD) and a C-terminal histidine phosphatase domain (PD) [5]. We recently found that the KD of both AtVIP enzymes can phosphorylate 5PP-InsP_5_ in vitro [38]. Additionally, the Hothorn group used NMR and showed that the product of the AtVIP enzymes is indeed 1,5-PP-InsP_4_ [37].

A second class of enzymes, known as the inositol hexakisphosphate kinases (IP6Ks), function in non-plant organisms by phosphorylating the 5-position of InsP_5_ and InsP_6_ to generate InsP_7_ [53]. While the genes coding for IP6Ks are present in humans and yeast, there is no identifiable *IP6K* gene in the plant genome [5]. This prompted us and other groups to speculate the AtVIPs might be bifunctional enzymes, phosphorylating both InsP_6_ and InsP_7_. While our biochemical analyses of the AtVIP KDs did not rule out the possibility that these enzymes can phosphorylate InsP_6_, they suggested that other enzymes likely had to exist in plants to drive this reaction [38]. As the ITPKs phosphorylate the 5-position of lower InsPs, we decided to target this class of enzymes, and found that both AtITPK1 and AtITPK2 are able to phosphorylate InsP_6_ in vitro [36,38]. We also demonstrated that the AtITPK1 product could be further phosphorylated by the AtVIP1-KD, resulting in InsP_8_ [38]. Based on our findings, as well as recent work by Laha et al., we conclude that the AtITPKs are the missing enzyme in the pathway [36,38].

## 5. How Do PP-InsPs Function in Plants?

Williams et al. suggested, in a review published in 2015, that a major function of PP-InsPs in plants was as a “glue” to bring together various protein binding partners [20]. At the time of the review, it was known that InsP_6_ could bind to the transport inhibitor response 1 (TIR1) auxin receptor [54]. Additionally, InsP_7_ was hypothesized to bind to the jasmonate (JA) receptor based on structural modeling experiments [34]. Exciting data, of importance to crop breeders, details how InsP_6_ and InsP_7_ are able to complex with key proteins involved in P*i*-sensing in plants [55]. Soils depleted in phosphorus lead plants to induce a suite of molecular and physiological mechanisms to enhance P*i* scavenging, known as the P*i* starvation response (PSR) [56,57]. The PSR is facilitated by an increase in transcription of a group of PSR genes, leading to increases in P*i* transport and uptake. Upregulation of PSR gene expression is regulated by Phosphate Starvation Response Regulator 1 (PHR1), a transcription factor, along with its homologs [56,57]. PHR1 and homologous transcription factors have a high binding affinity for promoters containing PHR1 binding sequences (P1BS), which allows for the binding and up-regulation of PSR genes under low P*i* conditions [56,57].

InsP_6_ and PP-InsPs regulate P*i* sensing via facilitating complex formations between the PHR1 transcription factor and the SPX domain-containing proteins (Figure 4) [58]. PHR and SPX proteins isolated from *Oryza sativa* (rice), known as OsPHR2 and OsSPX4, respectively, can complex with InsP_6_ or InsP_7_ in vitro [55]. InsP_8_ has an even lower dissociation constant than InsP_7_ in the OsPHR2–OsSPX4 complex formation [59]. Together, these data support the idea that InsP_8_ is the main mediator of the PHR1:SPX complex formation in plants. 

While InsP_7_ has a ~7-fold stronger binding affinity than InsP_6_ to OsSPX4, recent genetic analyses greatly support the idea that InsP_8_ is the major signaling molecule, or proxy, that conveys information on the P*i* status within the plant cell [37,61]. First, *atipk1*, *atitpk1*, *atitpk4* and *atvip1*/*vip2* mutants commonly show defects in P*i* sensing, such that the PSR is turned on even when grown under P*i*-replete conditions [30,33,37,61]. Additionally, all of these particular mutants have decreased InsP_8_ levels (Table 1). In the case of *atvip1*/*vip2* double mutants, the PSR is likely so highly upregulated that growth becomes stunted and lethality occurs in double homozygotes [37,61]. In contrast, *atipk1* and *atitpk1* mutants can grow fairly normally under certain conditions, and can be stimulated to further increase the PSR under P*i*-replete conditions [33,62]. All of this information suggests that InsP_8_ functions to turn off the PSR. 

We now know in greater detail that PP-InsPs likely function in binding to plant hormone receptors and transcription factor complexes involved in P*i* sensing. One example where PP-InsPs potentially function as cofactors is in the case of auxin signaling [54]. Auxin is a phytohormone which regulates numerous plant developmental processes and responses to environmental stress [63,64]. Auxin modulates gene expression by binding to the auxin receptor TIR1, an F-box protein, and mediates the SCF ubiquitin–ligase-catalyzed proteolysis of AUX/IAA transcriptional repressors [65,66]. The crystallized Arabidopsis TIR1 protein complex has a tightly bound InsP_6_ in the leucine-rich repeat (LRR) domain of TIR1, suggesting that InsP_6_ is a cofactor for the auxin receptor [54].

A later study identified Ins(1,2,4,5,6)P_5_ in the crystallized structure of a homologous plant hormone-JA co-receptor [67]. JA is a phytohormone critical for environmental and pathogen defense signaling as well as plant physiology [68]. Similar to auxin signaling with TIR1, JA is also perceived by interactions between F-box protein, coronatine-insensitive 1 (COI1), and the JA zim domain (JAZ) transcriptional repressors [69,70,71]. JA gene regulation is modulated by JA-hormone binding to COI1 and the degradation of JAZ repressors, freeing repressed transcription factors to upregulate JA genes [67]. Sheard et al. showed that Ins(1,2,4,5,6)P_5_ binds to and stabilizes COI, suggesting that Ins(1,2,4,5,6)P_5_ is a cofactor for the JA receptor [67,72]. A study using structural modeling predicted that the PP-InsPs have a much stronger binding affinity for COI1-JAZ1 than Ins(1,2,4,5,6)P_5_ and InsP_6_, suggesting that the PP-InsPs might be the true cofactor for the JA receptor [34]. 

## 6. Consequences for Targeting InsP_6_ for Reduction

With our current understanding of PP-InsP synthesis, and analyses of genetic mutants in the pathway, it seems reasonable to question whether reducing InsP_6_ in plants will also result in reduction in PP-InsPs. Most characterized Arabidopsis mutants showing alterations in InsP_6_ also have impacted the intracellular levels of InsP_7_ and InsP_8_ (Table 1). Mutants with reduced PP-InsPs, such as *atipk1*, *atitpk1*, and *atvip1*/*vip2* mutants, show an upregulation of the PSR, which could impact engineered crop performance in the field [33,37,61]. Specifically, altered P*i* sensing could negatively impact plant growth and development, reduce viability, and alter root architecture [56,57]. Alterations in P*i* sensing can affect other signaling systems, such as the sensing and accumulation of nitrate and other micronutrients, along with the ABA signaling pathway, often linked to stress pathways [73,74].

Given the link between InsP_8_ and JA signaling, decreasing PP-InsPs might result in crops that are more susceptible to pathogens and insects [34]. This impact is seen in Arabidopsis *atvip1* mutants, which have reduced InsP_8_, and show increased susceptibility to insect herbivory as compared to WT plants [34]. Additionally, both transgenic potato plants and Arabidopsis mutants with reduced *myo*-*inositol phosphate synthase* (MIPS), the first committed step in inositol synthesis, have decreased InsP_6_ and are more susceptible to pathogenic viruses [75]. While the PP-InsPs were not quantified in these mutants, it is possible that the PP-InsPs were also reduced and are a causative factor in the increased susceptibility to pathogens.

## 7. Concluding Remarks

Ultimately, understanding the genetic mechanisms for controlling PP-InsP synthesis and function is critical for developing novel low-phytate crops that are not compromised by changes in PP-InsP signaling. A recent review nicely details existing transgenic strategies used to reduce phytate in plants [76]. A common strategy is to use tissue-specific promoters to drive the overexpression of enzymes that break down InsP_6_, such as Phytase, or to knock-down the expression of genes required for InsP_6_ synthesis. By doing so, the idea is that only specific tissues will have reduced InsP_6_. This strategy works well to reduce InsP_6_ in seeds, for example, without negatively impacting vegetative tissues. One potential drawback of this approach is that the reduction in InsP_6_ may also affect the precursor available for PP-InsP synthesis in these transgenic plants. It is also important to consider that plants store approximately 1% InsP_7_ and InsP_8_ in seeds [5]. We do not know much currently about whether and how PP-InsPs might regulate seed phosphate storage, however, we point out that existing data on *mrp5* InsP_6_ transporter mutants indicate that InsP_6_ modulation in seed can result in increases in PP-InsPs [5]. Given the emerging role of PP-InsPs in controlling critical plant sensing and signaling pathways, the future development of strategies for phytate reduction without compromising PP-InsP synthesis and function should be considered by plant breeders.

## Figures and Tables

**Figure 1 plants-09-00115-f001:**
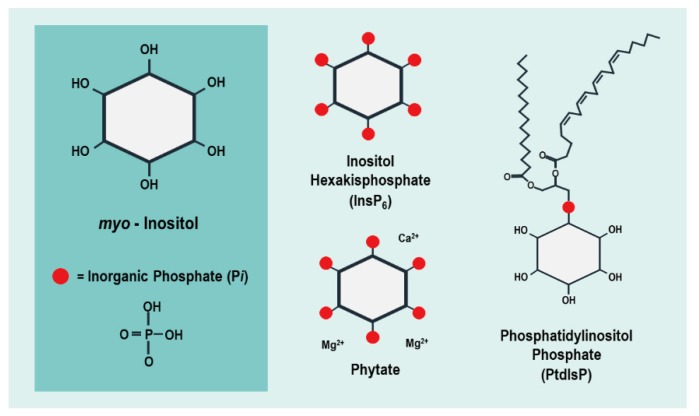
Simplified structures of *myo*-inositol, InsP_6_, phytate, and PtdInsP. Inorganic phosphate (P*i*) groups, covalently bound to the *myo*-inositol ring by an oxygen molecule, are represented in red. For simplification purposes, this figure does not take into account the axial and equatorial positions of the moieties.

**Figure 3 plants-09-00115-f003:**
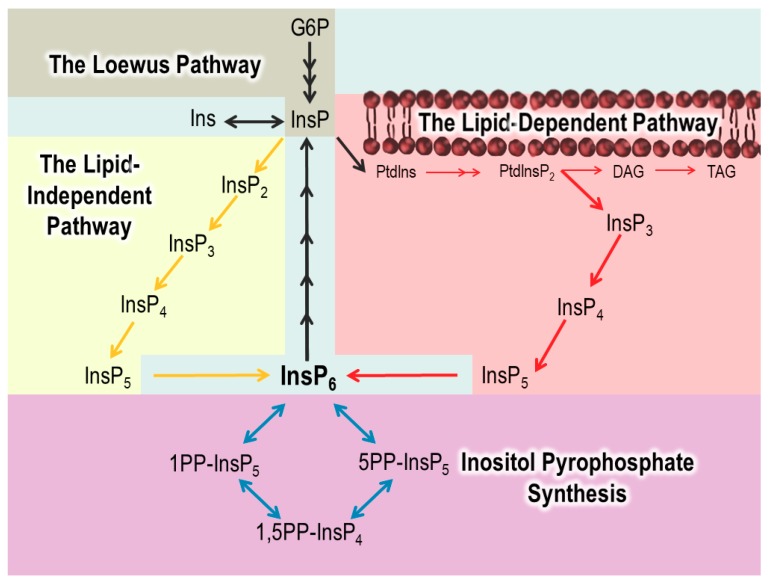
A simplified view of the InsP synthesis and degradation pathway. InsP synthesis starts in the Loewus Pathway (tan), where InsP is synthesized from Glucose-6-Phosphate (G6P). InsPs are synthesized through the Lipid-Dependent (pink) or Lipid-Independent (yellow) pathways. PP-InsPs in plants are synthesized from InsP_6_ (purple). The enzymes involved in the pathway are discussed throughout the review.

**Figure 4 plants-09-00115-f004:**
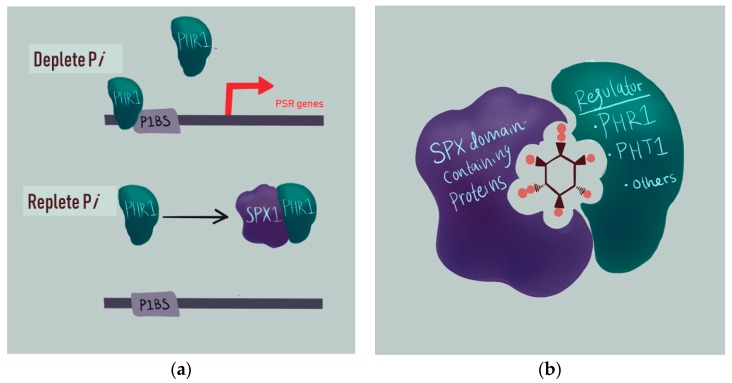
Model depicting how PP-InsPs regulate plant P*i* sensing and the P*i* starvation response (PSR). (**a**) PSR gene regulation under deplete (top) and replete P*i* conditions (bottom). SPX1 binds to PHR1 under replete P*i*, preventing SPX1 from binding to promoters containing P1BS. Under deplete P*i*, PHR1 is uninhibited from binding to P1BS promoters. Adapted from [60]. (**b**) Model depicting a complex formation between SPX1, PHR1, and InsP_8_. This model represents interactions between other SPX proteins and PSR regulators, such as PHR1 and its homolog, PHT1, along with others.

**Table 1 plants-09-00115-t001:** Loss-of-function Arabidopsis mutants and impacts on InsP_6_ and PP-InsP levels. Arabidopsis is a simple model system that can be used to gauge the impacts of genetic changes on InsPs. The table shows the impact on InsP_6_ and PP-InsPs in Arabidopsis mutants for enzymes involved in InsP synthesis. Mutants for enzymes important in both the Lipid-Dependent and Lipid-Independent pathways are indicated (*).

Pathway	Mutant	Impact on InsP_6_	Impact on PP-InsPs
**Lipid-Dependent Pathway**	*plc*	Nine genes; characterized single mutants have no change in InsP_6_ [32]	No Change [32]
	*ipk2α* *	Lethal Knock-Out [30]	Unknown
	*ipk2β* *	35% reduction in mass seed InsP_6_; no change in seedling tissue as measured by radiolabeling [30]	Unknown
	*ipk*1 *	83% reduction in mass seed InsP_6_; 93% reduction in seedlings as measured by radiolabeling [30]	InsP_7_ and InsP_8_ are reduced [33,34]
**Lipid-Independent Pathway**	*mik*	62–66% reduction in mass seed InsP_6_ [35]	Unknown
	*lpa*1	47–57% reduction in mass seed InsP_6_ [35]	Unknown
	*itpk*1–4	46% reduction in mass seed InsP_6_ in *Atitpk1* mutants; no changes in *Atitpk2* or *Atitpk3*; 40–51% reduction in *Atitpk4* [35]	*Atitpk1* and *Atitpk4* have reduced InsP_7_ and InsP_8_ [33,36]
**Phytate Storage**	*mrp*5	73–80% reduction in mass seed InsP_6_ [35]; decreased InsP_6_ in seeds and vegetative tissue as measured by radiolabeling [4]	Elevated InsP_7_ and InsP_8_ [5]
**PP-InsP Synthetic Pathway**	*vip*1	No change reported [34,37]	Increased InsP_7_ and Decreased InsP_8_ [34,37]
	*vip*1/*vip*2	No change reported [37]	Increased InsP_7_ and Decreased InsP_8_ [37,38]

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
