# Peer review of "Can Inositol Pyrophosphates Inform Strategies for Developing Low Phytate Crops?"

_plants, 2020, doi:10.3390/plants9010115_

Round 1
Reviewer 1 Report
The article entitled “Can Inositol Pyrophosphates Inform Strategies for Developing Low Phytate Crops?” discusses an important topic, the development of low phytate (IP6) crops, using an original and highly appropriate viewpoint, namely the inositol pyrophosphate (PP-IPs). The review is well articulated, initially describing inositol phosphates metabolic pathways and then the functions attributed to PP-IPs in plants. However, only in the final section, in the concluding remark, is the question expressed in the title addressed. The concluding remarks are fine as they are, but I was expecting more elaboration and speculation on the possible control of IP6 level in seeds by PP-IPs. For example, the possibility that better understanding of how PP-IPs control cellular phosphate homeostasis might lead to new strategies to reduce seeds IP6, ultimately a phosphate storage molecule, is completely missing. The review is overall well written, but there are important references missing and other places to improve.
- Line 14-16 Abstract. I suggest to change “inositol (1,3,4) triphosphate â…š kinase enzymes” to “ITPKs”. It is otherwise confusing for the inexpert reader to understand why an inositol trisphosphate kinase metabolises the conversion IP6 to IP7. Similarly, I also suggest changing “diphosphoinositol pentakisphosphate kinase” to “PPIP5K”.
- Line 119-149 Section 3. InsP6 Synthesis: The Lipid-Independent Pathway
The author recognises the importance of a very recent publication (December 3; Desfougères et al. Proc Natl Acad Sci U S A. 2019;116:24551-24561) describing the cytosolic route to IP6 synthesis and correctly cited this work. However, this looks to be a last minute addition. The authors should rewrite this section, properly integrating the work of Desfougères et al. (referencing it also in table 1), which demonstrates that ITPK (mammalian as well as plant enzymes) phosphorylate IP generated directly from glucose-6P by myo-inositol phosphate synthase or from inositol by the myo-inositol kinase (lpa3). It is important to mention and discuss that maize lpa2 corresponds to ITPK (Shi et al. Plant Physiol. 2003;131:507-15), and indicate that IPMK belongs to both IP6 biosynthetic pathways. The lipid-independent pathway is NOT a unique, plant-specific pathway. The cytosolic route was initially identified in amoeba (Dictyostelium discoideum) in 1990, and the landmark paper from Stephens and Irvine must be discussed and cited (Nature. 1990;346:580-3).
- Line 135-137 and 171-174. This article from Schaaf’s lab must be cited and discussed: Laha et al. ACS Chem Biol. 2019;14:2127-2133.
- Figure 3. I suggest redrawing this figure, giving the same “graphical weight” to both the lipid-dependent and -independent pathways. Currently the use of red arrows and larger pinkish section make the lipid-dependent pathway far more evident than the lipid-independent route. This graphic representation gives the impression that the lipid-dependent pathway is the more important of the two. Most likely, in plant the soluble route plays the major role in IP6 synthesis.
Author Response
.......For example, the possibility that better understanding of how PP-IPs control cellular phosphate homeostasis might lead to new strategies to reduce seeds IP6, ultimately a phosphate storage molecule, is completely missing.
We understand this point and had originally considered detailing potential specific strategies in this review. Apart from the issue that this would be seen as too speculative at this point, we were also worried that a critical understanding of such strategies would require specialized knowledge of the inositol pyrophosphate pathway. So we focused on providing tknowledge about the inositol pyrophosphates, and the pathways and functions, known at this time. We hope that this treatment will be assessible to crop breeders interested in InsP6, and get the inositol pyrophosphates on their “radar screens”.
- Line 14-16 Abstract. I suggest to change “inositol (1,3,4) triphosphate â…š kinase enzymes” to “ITPKs”. It is otherwise confusing for the inexpert reader to understand why an inositol trisphosphate kinase metabolises the conversion IP6 to IP7. Similarly, I also suggest changing “diphosphoinositol pentakisphosphate kinase” to “PPIP5K”.
This change was made in the abstract.
- Line 119-149 Section 3. InsP6 Synthesis: The Lipid-Independent Pathway
The author recognises the importance of a very recent publication (December 3; Desfougères et al. Proc Natl Acad Sci U S A. 2019;116:24551-24561) describing the cytosolic route to IP6 synthesis and correctly cited this work. However, this looks to be a last minute addition. The authors should rewrite this section, properly integrating the work of Desfougères et al. (referencing it also in table 1), which demonstrates that ITPK (mammalian as well as plant enzymes) phosphorylate IP generated directly from glucose-6P by myo-inositol phosphate synthase or from inositol by the myo-inositol kinase (lpa3).
Reviewer 1 has correctly identified that the Desfougères et al paper was a last minute addition to our review! This highly impactful article had become available to us a few days before the review submission deadline. In this revised manuscript, we feel we have properly integrated this work. We first mention this work on p. 3, and point out that it is a crucial paper on understanding the evolution of the enzymes in the pathway. On p. 6 we mention this work again, and describe how the gene restores PLC mutants in production of InsP6, and the implications of this. It had also been requested that we integrate the work by Desfougères et al. into Table 1. We tried to do this, however, it became confusing as Table 1 describes the impact on InsP6-8 levels in genetic mutants, Arabidopsis specifically. We feel this would only confuse readers.
- Line 135-137 and 171-174. This article from Schaaf’s lab must be cited and discussed: Laha et al. ACS Chem Biol. 2019;14:2127-2133.
We have incorporated the important work done by Laha et al (2019) in the appropriate sections suggested.
It is important to mention and discuss that maize lpa2 corresponds to ITPK (Shi et al. Plant Physiol. 2003;131:507-15), and indicate that IPMK belongs to both IP6 biosynthetic pathways. The lipid-independent pathway is NOT a unique, plant-specific pathway. The cytosolic route was initially identified in amoeba (Dictyostelium discoideum) in 1990, and the landmark paper from Stephens and Irvine must be discussed and cited (Nature. 1990;346:580-3).
We have discussed the lpa2 maize mutant in Shi et al. on p. 5, and indicated that IPMK belongs to both InsP6 pathways (p. 6). We have corrected our discussion of the Lipid-Independent pathway, and clarified where discovery of the pathway was made. Thus we added the Stephens and Irvine work, along with work from the Brearley lab.
- Figure 3. I suggest redrawing this figure, giving the same “graphical weight” to both the lipid-dependent and -independent pathways. Currently the use of red arrows and larger pinkish section make the lipid-dependent pathway far more evident than the lipid-independent route. This graphic representation gives the impression that the lipid-dependent pathway is the more important of the two. Most likely, in plant the soluble route plays the major role in IP6 synthesis.
We did not intend to slight the Lipid-independent pathway by making it graphically smaller! We have re-colorized and formatted this figure so that both the Lipid-Independent and -Dependent pathways have equal weights and that one pathway does not appear to be more important than the other.
Reviewer 2 Report
The manuscript is very well written and organized. I have only some minor suggestions:
Line 48: I am not sure that the verb "serve" is properly used in this sentence.
Line 57-58: the sentence does not properly sound.
Line 61: maize and corn are the same plant, please revise.
Table 1: if authiors would like to refer to mutants they have to write all the names in the second column in lowercase letters.
The Arabidopsis mutants are normally indicated with "at" written in lowercase letters. Please, revise.
Line 126: Please substitute with "Lipid-Independent"
Author Response
Reviewer 2
The manuscript is very well written and organized. I have only some minor suggestions:
Line 48: I am not sure that the verb "serve" is properly used in this sentence.
The word serve was changed.
Line 57-58: the sentence does not properly sound. Line 61: maize and corn are the same plant, please revise.
Figure legend 2 was re-worded and the maize/corn issue was resolved.
Table 1: if authiors would like to refer to mutants they have to write all the names in the second column in lowercase letters.
These were changed to lowercase.
The Arabidopsis mutants are normally indicated with "at" written in lowercase letters. Please, revise.
We revised several mentions in the text using the at, as requested.
Line 126: Please substitute with "Lipid-Independent"
This change was made.
Reviewer 3 Report
This manuscript provides an overview of PP-InsP synthesis/signaling and elucidate the strategies for developing crops with low phytate content. The manuscript is overall well structured and written and relevant literature is cited. Based on my comments I recommend the ms to be accepted in present form.
Author Response
Reviewer 3 had no concerns.